# Health impact of hepatic-venous-occlusive disease in a small town in Ethiopia—Case study from Tahtay koraro district in Tigray region, 2017

Kissanet Tesfay Weldearegay[1]*, Mekonnen Gebremichael Gebrekidan[2], Alefech Adissu Gezahegne[1]

1 Mekelle University College of Health Science School of Public Health, Department of Epidemiology, Mekelle, Ethiopia, 2 Tigray Regional Health Bureaus, Mekelle, Ethiopia

☯ These authors contributed equally to this work.
* tesfaykissanet@gmail.com

## Abstract

**Data Availability Statement:** All relevant data are within the paper and its Supporting Information files.

### Background

Hepatic venous-occlusive disease is blockage of microscopic veins in the liver causing 20–50% mortality. Ingestion of pyrrolizidine alkaloid plant, radiation therapy, and post-bone-marrow-transplant reactions are the commonest causes. In Ethiopia, a venous-occlusive disease outbreak was identified in 2002 in Tahtay Koraro district, Tigray. Suspected due to ingestion of the toxic pyrrolizidine alkaloid plant *Ageratum conyzoids*, found throughout the district. We aimed to describe the surveillance data of venous-occlusive disease from September 2006 to August 2016 in Tahtay koraro district, Ethiopia, 2017.

### Methodology

We defined a possible Hepatic venous-occlusive disease case as any patient with abdominal pain for at least 2 weeks, abdominal distention, and hepato-splenomegaly during September 2006-August 2016. We reviewed previous district line lists, weekly reports, and clinical records to identify and describe cases. Agricultural interventions were obtained from the agricultural offices of the district.

### Result

We identified 179 possible cases with 83 deaths with a case-fatality rate of 46.3%. Among cases, 110 (61.5%) were males and 113 (63%) were >15 years. In total, 164 (91.6%) cases were from one village (Kelakil). The pick number of cases of VOD in this village was during 2008/09 which was 1076. The highest incidence (86/100,000) occurred in 2008. During the study period, 2,746 years of potential life were lost due to Hepatic venous-occlusive disease. Mechanical removal of the *Ageratum* started in 2011.

**Funding:** The author(s) received no specific funding for this work.

**Competing interests:** The authors have declared that no competing interests exist.

**Abbreviations:** CFR, Case Fatality Rate; PA, Pyrrolizidine Alkaloid; VOD, Venous-Occlusive Disease; YPLL, Years of Potential Life Lost.

## Conclusion

Hepatic venous-occlusive disease was an ongoing problem in Tahtay Koraro; However, the problem has largely been alleviated by displacing people from the affected area and removing the causative weed. More research is needed to understand why Kelakil village was more affected despite the widespread presence of the weed. Chemical and mechanical removal of the *Ageratum* could strengthen intervention activities.

## Introduction

The venous-occlusive disease of the liver (VOD) is blockage of the very small veins in the liver. Because flow out of the liver is blocked, blood backs up in the liver. VOD is thought to be caused by damage to sinusoidal endothelial cells and hepatocytes in zone three of the liver [1–5].

Common causes include; Ingestion of pyrrolizidine alkaloids (PAs), use of certain drugs that occasionally have toxic effects on the liver including cyclophosphamides and azathioprine, radiation therapy, a reaction after bone marrow or stem cell transplantation, contraceptive and, several antineoplastic drugs [4, 6, 7].

Around 3% of the world's flowering plants have one or more of the toxic PAs. The cause of VOD can be carelessly harvested wild weed with grain crops and consumption of flour contaminated by these plants [6]. Consumption of PAs-containing food or plants not necessarily causes signs of VOD. Because it depends on the dose consumed [8].

Sign of symptoms of VOD is characterized by the triad of weight gain caused by fluid retention, tender hepatomegaly and hyperbilirubinemia [4, 7, 9]. Diagnosis is conducted usually based on signs and symptoms, having ruled out other conditions that can mimic the disease [9]. Confirmation diagnosis is often conducted using Doppler ultrasonography [4]. Histological biopsy of the liver is the gold standard for the diagnosis of VOD [8].

Two-thirds of patients who died have multi-organ failure, this causes difficulty in knowing the main cause of death. In different kinds of literature, it has been reported that the incidence of VOD varies from 0 to 70% and its mortality from 20 to 50% [6].

In Western Afghanistan, Gulran District from 1974 to 1976 the largest outbreak of VOD was recorded causing an estimated 7800 people with about 1600 deaths with a case fatality rate of 20.5%[10].

In November-December 1975, in Sarguja district of India; the VOD outbreak was reported due to consumption of cereals mixed with seeds of the plant (*Crotalaria* sp.) containing pyrrolizidine alkaloids [11].

In northern Iraq in 1998, an outbreak of VOD resulted in 14 patient admissions to Mosul hospital the entire patient had VOD of the liver as confirmed by histopathological examination. The underlying causative agent was isolated from the contaminated wheat and flour and found to be pyrrolizidine alkaloids [1].

In Ethiopia, a liver disease of unknown etiology, called the unknown liver disease by the community, was first identified in 2002 in kelakil kebelle, Tahtay koraro district, northwestern Tigray; a rugged, semi-arid, mountainous region [12]. From that area sacrificed animal liver on histopathological examination test result showed severe hepatic necrosis. The serum sample test result indicated raised level of some clinical markers that are highly significant for detecting liver abnormality of toxic origin [13]. On analysis of risk factors for the cause showed that unprotected drinking water source was associated with the VOD [14, 15]. The aim of this

study is to assess the surveillance data of VOD from September 2006 to August 2016 in Tahtay koraro district, North West Zone of Tigray region, Ethiopia, 2017.

## Methods

### Study area and period

Tahtay koraro district is found at altitude & latitude 140 06' 00"north 380 16' 00"meters above & below the sea level. The climatic condition of the district is 2% high land, 75% semi-lowland and, 23% lowland. The annual temperature is estimated to be between 15˚c to 25˚c. The annual rainfall range is 726–1400 mm. In 2017, the district's total population was 78,311 [16]. The study was conducted from February to March 2017 in Tahtay Koraro district, Tigray Region of Ethiopia.

### VOD standard case definitions

**Suspected VOD case**: is defined as a person with abdominal distention and either a household member sick with similar symptoms or abdominal cramp/pain for at least two weeks [17].

**Possible VOD case**: is defined as a person who meets the suspected case definition and has hepatomegaly or splenomegaly [17].

**Probable VOD case**: is defined as a person who meets the possible case definition and has a serum alkaline phosphatase greater than or equal to twice the upper limit of the normal [17].

### Study design

A cross-sectional study was conducted.

### Source population

Populations of Tahtay koraro district from 2006 to 2016.

### Study subjects

All Possible and Suspected VOD cases and deaths registered on line list of Tahtay koraro district from September 2006 to August 2016.

### Study units

A person who has the disease in the analysis period.

### Data collection and analysis

Using structured checklist VOD cases of the last ten consecutive years were reviewed and collected from the line lists and weekly reports of the district health office. Data was reviewed, cleaned, entered and analyzed using Microsoft Excel.

### Ethics approval and consent to participate

Ethical clearance was obtained from an ethical review board of Mekelle University, College of Health Sciences and Department of Public Health. A letter of permission was written from the Tigray Regional Health Bureau to Tahtay Koraro district health office. The study was done after permission was obtained from Tahtay Koraro district health office. Informed consent was waived by ethical review board. The confidentiality of information regarding patients involved in this study was maintained by avoiding identifying study participants by name.

## Results

In Tahtay koraro district a total of 179 Hepatic VOD cases and 83 deaths were reported during the study time. The case fatality rate (CFR) was 46.3%. Males accounted for 110(61.5%) of the VOD cases.

All the cases were diagnosed without laboratory confirmations. Of the total cases, 88 (49.72%) were possible cases and 54(30.17%) were suspected cases, and 37(20.7%) were not properly registered.

The median age of cases was 29, ranges from 1 to 78 Years. When we see the age distribution, <5 years of age 14(8%), 5–14 years of age 52(29%) and above 15 years of age group accounted 113(63%) (Fig 1).

Among the total 83 deaths, 57(69%) were males. The median ages of the VOD deaths were 30.5. Of the total deaths above 15 years of age accounted for 58(70%), 5–14 years of ages were 23(28%), and less than 5 years were 2(1.1%). During the ten years, 2746 years of potential life lost (YPLL) were attributed due to VOD. During the ten years period, there was no female death among less than five years of children's. While 17(65.4%) females and 41(72%) males' deaths occurred among the age group of above 15 years (Fig 2).

During the ten years period of analysis, Of all cases, 164 (91.6%) were occurred in Kelekil kebelle, while the other kebelles of Tahtay koraro contributed less than eight percent of the total cases. From the total cases in Kelakil kebelle, 172(96%) cases have occurred in Tsaeda emba sub kebelle.

In the district, the highest incidence rates of VOD cases were 86 per 100,000 populations from September 2008 to August 2009. While the lowest incidence rates were 3 cases per 100,000 populations from September 2015 to August 2016.

The highest affected Kebelle of the district was Kelakil with the highest incidence rate of 1076 cases from September 2008 to August 2009 and the lowest 33 cases from September 2015 to August 2016 per 100,000 populations. It was the only kebelle which had the disease throughout the ten years of analyzed data. (Table 1).

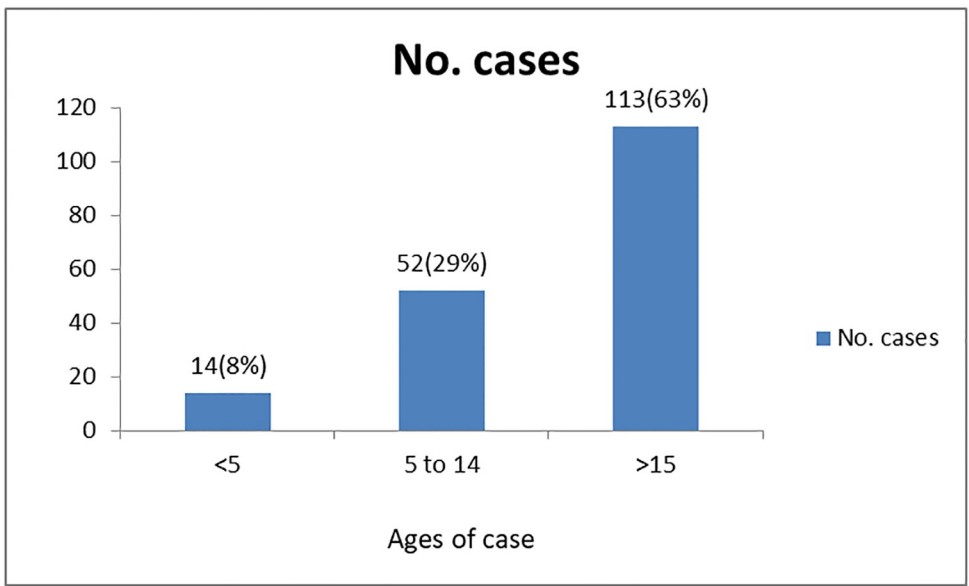

**Fig 1. HVOD case by age categories in Tahtay koraro district from September 2006 to August 2016.**

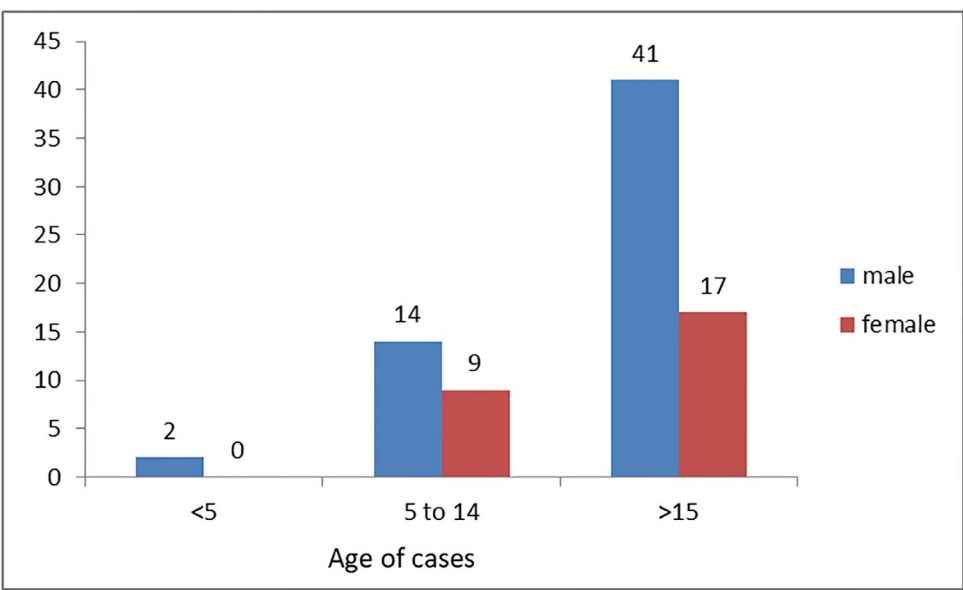

**Fig 2. The proportion of HVOD deaths by sex and age, in Tahtay koraro during 2006–2016.**

The incidence rate of the cases started to decrease dramatically from 86 cases in 2008/09 to 19 cases in 2009/10 in the district and from 1076 cases in 2008/09 to 242 cases in 2009/10 in Kelakil kebelle per 100,000 populations, since then the disease was started to decrease and in 20015/16 the incidence rate was only 3 and 33 cases per 100,000 populations in the district and in Kelakil kebelle respectively. As the incidence rate in the kebelle decreases, the incidence rate in the district also decreases (Fig 3).

The prevalence of VOD in the district in 2015/16 was 3 cases per 1000 populations while in Kelakil kebelle 34 cases per 1000 populations.

Out of the 83 deaths, 79(95%) deaths occurred in Kelakil kebelles while Tahtay adigebaro accounted for two deaths and Beles and lemlem kebelles accounted for one case each. The case fatality rate of VOD among males was 52%(57/110) while among females was 38%(26/69). In Kelakil kebelle the case fatality rate was 48% (79 out of 164). In the district, the case fatality rate

**Table 1. Ten year's incidence rates of HVOD case in Tahtay koraro district, during 2006–2016.**

| Sr.no | years | The incidence rate in sub kebelles per 100,000 population | | | | | | | Incidence of the district per 100,000 population |
|---|---|---|---|---|---|---|---|---|---|
| | | Kelakil | Beles | Tahtay adigebru | Lemlem | adigidad | adimenabir | mydemu | |
| 1 | 2006/07 | 919 | 0 | 0 | 0 | 0 | 0 | 0 | 74 |
| 2 | 2007/08 | 936 | 0 | 0 | 0 | 0 | 0 | 0 | 75 |
| 3 | 2008/09 | 1076 | 0 | 0 | 0 | 0 | 0 | 0 | 86 |
| 4 | 2009/10 | 242 | 33 | 123 | 48 | 0 | 0 | 0 | 19 |
| 5 | 2010/11 | 195 | 0 | 60 | 0 | 41 | 119 | 0 | 15 |
| 6 | 2011/12 | 94 | 0 | 0 | 0 | 0 | 0 | 0 | 8 |
| 7 | 2012/13 | 55 | 0 | 0 | 0 | 0 | 0 | 0 | 4 |
| 8 | 2013/14 | 105 | 0 | 0 | 0 | 0 | 0 | 0 | 8 |
| 9 | 2014/15 | 51 | 0 | 0 | 0 | 0 | 0 | 0 | 4 |
| 10 | 2015/16 | 33 | 0 | 0 | 0 | 0 | 0 | 14 | 3 |

NB. The first year started in September and ends in August of the consecutive year.

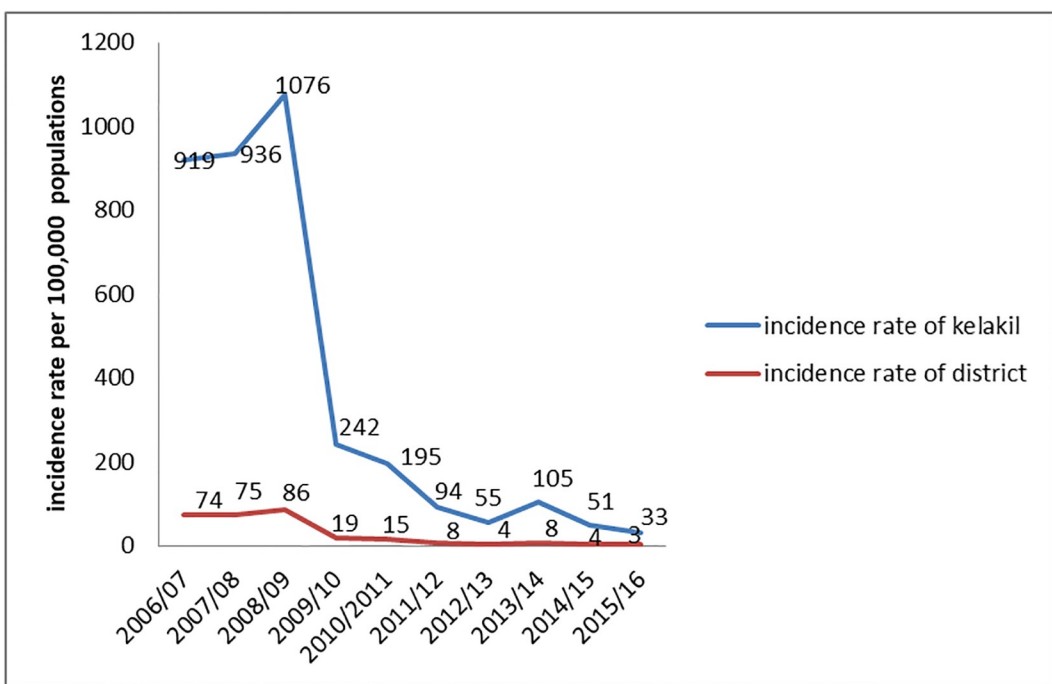

**Fig 3. Ten-year incidence rate of Tahtay koraro district and kelakil kebelle, Tigray, from 2006 to 2016.**

of VOD was started to decrease gradually, from 17 (40.5%) in 2006/07 to 2 (1.9%) in 2015/16 (Fig 4).

## Discussion

In 2008 in the district, the highly affected Tsaedaemba sub kebelle residents were displaced to the lowly affected Endabanou sub kebelle within the Kelakil kebelle for decreasing the incidence and mortality of the disease.

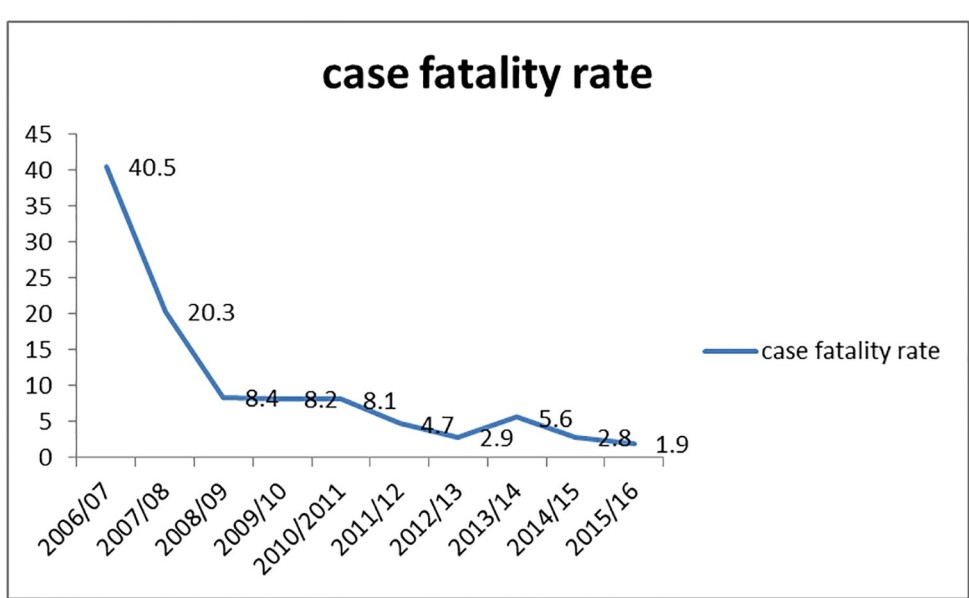

**Fig 4. Trends of HVOD by case fatality rate, in Tahtay koraro district, from 2006 to 2016.**

Even though the district had not organized data regarding intervention activities of unweeding of the ageratums since 2011 Kelakil kebelle residents started unweeding of the ageratum. In 2016 un weeding of the ageratum have been started in all kebelles of the district, and Out of 9092 hectares to being unwedded, mechanically unweeding of the ageratum was done in July and September, with 8099(89%) and 8829(97%) of hectares respectively [16].

Our study finding conducted from September 2006 to August 2016 in Tahtay koraro district showed that, out of a total of 179 VOD cases, 110 (61.5%) were males. Similarly the analysis (from 2002 to 2012) of a study conducted in northwestern Tigray of Ethiopia showed that; out of total 1095 VOD cases, 766(70%) were males. And another outbreak study of VOD happened in 1998 in northern Iraq showed that; out of the 14 patient 8(57%) of them were males [6, 17]. The studies have indicated that males were highly affected. The reason might be due to alcohol consumption is higher in males than females and alcohol has the ability to enhance the enzyme Cytochrome P450 which is responsible for the conversion of nontoxic PAs to toxic metabolites in the liver.

The median age of the cases was 29 years, which was similar to the study conducted in northwestern Tigray which was 30 years [17]. In our study, the minimum and maximum age ranges of cases were 1 to 78 years. This was almost the same as the study conducted in northern Tigray, which ranges 1 to 81 years. The reason for one year's old child case might be due to PAs has been found in human milk during PA poisoning epidemics. So that children's might have the disease during breastfeeding. From this finding, we understand that VOD can affect all age categories. Our study showed that; the age groups above 15 years were more affected 52 (29%). This is similar compared with a study conducted in northwestern Tigray indicated that age group above 15 years was the dominant affected group [17]. The reason might be, adults are more participated in drinking alcohol and eating raw meats like animal liver, so that they were more at risk getting off the disease.

Majority of the cases 164 (91.6%) were in Kelakil kebelle of the district. This might be due to the reason that the residents were consuming more PAs than others kebelles. Also, it could be due to the reason that the area might have high acid in the soil which decreases the organic matters of the soil and enhancing the growth of the ageratum weeds. If the density of ageratum weed is high, obviously VOD cases will be increased. And also the kebelle might be less involved in an unweeding of the ageratum before and after harvest than the other kebelles.

In this study, the incidence rate started to decrease dramatically from 86 cases in 2008/09 to 19 cases in 2009/10. In Kelakil kebelle of the district, there was decrement of cases per 100,000 populations from 1076 in 2008/09 to 242 in 2009/10. This decrement might be due to the displacement of residents from Tsaedaemba sub kebelle which were highly affected area to lowly affected area Endabanou sub kebelle within kelakil kebelles in 2008. Since then the disease was decreasing gradually and in 20015/16 the incidence rate was 3 and 33 cases per 100,000 populations in the district and in Kelakil kebelle respectively. In addition to the above reason, the decrement might be due to the intervention activities done in 2011. Activities such as unweeding of ageratum weed and health education activities given to the residents on the importance of weeding crops by health extension workers. And also there might be under-reporting of cases especially after the finding of the causative agent.

During the analysis period of our study, the district's case fatality rate of hepatic VOD was 46.4% (83 out of 179 cases). Which is more than two times higher than an outbreak happen in a study done in, Western Afghanistan Gulran District 1974–1976, with a case fatality rate of 20.5%(1600 out of 7800). And also it was higher compared with analysis conducted in northwestern Tigray region from 2002–2012, which was 27.6% (302/1095), whereas it is almost similar to findings from India, Sarguja district, 1975 with case fatality rate of 42% (28 out of 67 cases) [10, 17].

In this study, the case fatality rate of VOD among males was 51.8%(57/110) and in females 37.7%(26/69). This result disagrees with the result of northwestern zone of Tigray 25.6% (196/766) among males and 32.2% (106/329) in females.

Our study reveled that, the case fatality rate of VOD was started to decrease gradually from 17(40.5%) in 2006/07 to 2 (1.9%) in 2015/16. The reason might be due to the improvement of the health professional's capacity for clinical care and management.

## Conclusion

A VOD case with a high case fatality rate was identified in Tahtay Koraro District, northwestern Tigray of northern Ethiopia during the ten years of analysis. Cases and case fatality rates were higher among males. The above 15 years of age group were the dominant group affected. The majority of the cases were from Kelakil Kebelle of the district. In the district, the highest incidence rate of VOD cases was occurred in 2008/09, while the lowest was in 2015/16. After the displacement of the highly affected villages, intervention activities of unweeding of the ageratum weed, and proper case management started and cases fatality rate of VOD decreased significantly. More research is needed to understand why Kelakil village was more affected despite the widespread presence of the weed. Chemical and mechanical removal of the *Ageratum* could strengthen intervention activities.

## Limitation

There was no complete registered data of some variables. Even though there were 83 deaths, it was difficult being sure that the causes of the deaths were only VOD, there might be coincidental events or other factors during the time because the death of the report was community-based.

## Supporting information

**S1 File. Data available.**
(SAV)

## Acknowledgments

We would like to thank Mekelle University, College of Health Science, Department of Epidemiology, and Tahtay Koraro district health office staff. Finally, we would like to thank all our family and friends for their essential support.

## Author Contributions

**Data curation:** Kissanet Tesfay Weldearegay, Mekonnen Gebremichael Gebrekidan.

**Methodology:** Kissanet Tesfay Weldearegay, Mekonnen Gebremichael Gebrekidan, Alefech Adissu Gezahegne.

**Project administration:** Alefech Adissu Gezahegne.

**Supervision:** Kissanet Tesfay Weldearegay, Mekonnen Gebremichael Gebrekidan.

**Writing – original draft:** Kissanet Tesfay Weldearegay, Mekonnen Gebremichael Gebrekidan, Alefech Adissu Gezahegne.

**Writing – review & editing:** Kissanet Tesfay Weldearegay, Mekonnen Gebremichael Gebrekidan.

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
