## [Decision Letter · Decision Letter 0]

19 Aug 2019

PONE-D-19-19407

Health impact of hepatic-venous occlusive disease in Tahtay Koraro district - Tigray region, Ethiopia, 2017

PLOS ONE

Dear Mrs Weldearegay,

Thank you for submitting your manuscript to PLOS ONE. After careful consideration, we feel that it has merit but does not fully meet PLOS ONE’s publication criteria as it currently stands. Therefore, we invite you to submit a revised version of the manuscript that addresses the points raised during the review process.

The manuscript is potentially interesting provided the authors are willing to address reviewers' concerns.

We would appreciate receiving your revised manuscript by Oct 03 2019 11:59PM. To enhance the reproducibility of your results, we recommend that if applicable you deposit your laboratory protocols in protocols.io, where a protocol can be assigned its own identifier (DOI) such that it can be cited independently in the future. For instructions see: http://journals.plos.org/plosone/s/submission-guidelines#loc-laboratory-protocols

We look forward to receiving your revised manuscript.

Kind regards,

Raffaele Serra, M.D., Ph.D

Academic Editor

PLOS ONE

Journal Requirements:

2. In ethics statement in the manuscript and in the online submission form, please provide additional information about the patient records/samples used in your retrospective study. We note taht you mention that you "obtained written informed consent from the study participants". However, we note that some participants had died by the time your study was conducted. Please therefore ensure that you have discussed whether all data were fully anonymized before you accessed them and/or whether the IRB or ethics committee waived the requirement for informed consent. If patients provided informed written consent to have data from their medical records used in research, please include this information.

3. We noticed you have some minor occurrence of overlapping text with the following previous publication(s), particularly in the Introduction, which needs to be addressed:

http://njmonline.nl/getpdf.php?id=933

https://doi.org/10.1016/S0973-6883(11)60128-X

https://doi.org/10.1016/0268-960X(93)90023-W

http://dx.doi.org/10.1155/2010/313280

In your revision ensure you cite all your sources (including your own works), and quote or rephrase any duplicated text outside the Methods section. Further consideration is dependent on these concerns being addressed.

4. We noticed you have some minor occurrence of overlapping text with the following previous publication(s), particularly in the Introduction, which needs to be addressed:

http://njmonline.nl/getpdf.php?id=933

https://doi.org/10.1016/S0973-6883(11)60128-X

https://doi.org/10.1016/0268-960X(93)90023-W

http://dx.doi.org/10.1155/2010/313280

In your revision ensure you cite all your sources (including your own works), and quote or rephrase any duplicated text outside the Methods section. Further consideration is dependent on these concerns being addressed."""

6. Please ensure that you refer to Figures 1-4 in your text as, if accepted, production will need this reference to link the reader to the figure.

7. We note you have included a table to which you do not refer in the text of your manuscript. Please ensure that you refer to Table 1 in your text; if accepted, production will need this reference to link the reader to the Table.

Additional Editor Comments:

The manuscript needs extensive revision according to reviewers' commnents.

Reviewers' comments:

Reviewer's Responses to Questions

**Comments to the Author**

1. Is the manuscript technically sound, and do the data support the conclusions?

Reviewer #1: Yes

2. Has the statistical analysis been performed appropriately and rigorously? 

Reviewer #1: Yes

3. Have the authors made all data underlying the findings in their manuscript fully available?

Reviewer #1: No

4. Is the manuscript presented in an intelligible fashion and written in standard English?

Reviewer #1: Yes

5. Review Comments to the Author

Reviewer #1: The study titled 'Health impact of hepatic-venous occlusive disease in Tahtay Koraro district- Tigray region, Ethiopia, 2017 is a unique study that reports on the epidemiology of the disease in a small town in Ethiopia.Hepatic venous occlusive disease seems to be uniquely affecting this small village and the risk factors are not known. There is very scant information on the impact as well as epidemiology of the disease and hence the study is significant. However, the manuscripts needs some editing before it can be published.

1. the title could be modified to "Health impact of hepatic-venous occlusive disease in a small town in Ethiopia- case study from Tahtay Koraro district in Tigray region, 2017

2. the results in the abstract should describe the trend over time of VoD fatality given in figure 3. They should describe the change points in 2008/09 and 2009/10 and why these inflection years.

3. in the abstract, line 29, after however, they should add 'the problem...

4. on page 3, line 53, instead of saying 'according to ...., it is sounds better to say, 'It has been reported that the incidence ...

5. on page 4, line 69, they should add a couple of sentences that describe what the current gaps are what the study adds to the current literature. it looks like that there are only two studies (ref 12 and 14) that have been made on this disease but none of them are published. But, searching the internet with "hepatic-venous occlusive disease in Tahtay Koraro district- Tigray region" came up with may more studies that should be included in this review. For example, I listed 7 below (and there may be more)

0. Cindy Chiu,1 Colleen Martin,1 Daniel Woldemichael,2 Girmay W Selasie,2 Israel Tareke,3 Richard Luce,4 Gidey G Libanos,5 Danielle Hunt,1 Tesfaye Bayleyegn,1 Adamu Addissie,6 Danielle Buttke,1 Amsalu Bitew,7 Sara Vagi,1 Matthew Murphy,1 Teshale Seboxa,6 Daddi Jima,8 and Asfaw Debella8. SURVEILLANCE OF A CHRONIC LIVER DISEASE OF UNIDENTIFIED CAUSE IN A RURAL SETTING OF ETHIOPIA: A CASE STUDY. Ethiop Med J. 2016 Jan; 54(1): 27–32.

1. Bane A, Seboxa T, Mesfin G, Ali A, Tsegaye Y, M WT, et al. An outbreak of veno-occlusive liver disease in northern Ethiopia, clinical findings. Ethiopian medical journal. 2012 Apr;50(Suppl 2):9–16. PubMed PMID: 22946291. Epub 2012/09/06. eng. [PubMed] [Google Scholar]

2. Mesfin G, Ali A, Seboxa T, Bane A, Tensae MW, Gebressilassie S, et al. An epidemiological study into the investigation of liver disease of unknown origin in a rural community of northern Ethiopia, 2005. Ethiopian medical journal. 2012 Apr;50(Suppl 2):1–8. PubMed PMID: 22946290. [PubMed] [Google Scholar]

3. Schneider J, Tsegaye Y, M WT, S GS, Haile T, Bane A, et al. Veno-occlusive liver disease: a case report. Ethiopian medical journal. 2012 Apr;50(Suppl 2):47–51. PubMed PMID: 22946295. [PubMed] [Google Scholar]

4. Debella A, Abebe D, Tekabe F, Mamo H, Abebe A, Tsegaye B, et al. Toxicity study and evaluation of biochemical markers towards the identification of the causative agent for an outbreak of liver disease in Tahtay Koraro Woreda, Tigray. Ethiopian medical journal. 2012 Apr;50(Suppl 2):27–35. PubMed PMID: 22946293. [PubMed] [Google Scholar]

5. Abebe D, Debella A, Tekabe F, Mekonnen Y, Degefa A, Mekonnen A, et al. An outbreak of liver disease in Tahtay Koraro Woreda, Tigray region of Ethiopia: a case-control study for the identification of the etiologic agent. Ethiopian medical journal. 2012 Apr;50(Suppl 2):17–25. PubMed PMID: 22946292. [PubMed] [Google Scholar]

6. Debella A, Abebe D, Tekabe F, Degefa A, Desta A, Tefera A, et al. Physico-chemical investigation of consumables and environmental samples to determine the causative agent of liver disease outbreak in Tahitay Koraro Woreda, Tigray. Ethiopian medical journal. 2012 Apr;50(Suppl 2):37–45. PubMed PMID: 22946294. [PubMed] [Google Scholar]

6. On page 4, lines 72- please remove this line.

7. On page 4, line 76, please add population size.

8. the results section needs some editing with each table and figure described fully. Start as follows:

Table 1 shows ....

9. There are many typographical errors that need to be edited. The corresponding author's e-mail is misspelled, on page 5, line 97, Excel is misspelled.

6. PLOS authors have the option to publish the peer review history of their article (what does this mean?). If published, this will include your full peer review and any attached files.

Reviewer #1: Yes: Mulugeta Gebregziabher, PhD

---

## [Author Response · Author response to Decision Letter 0]

3 Oct 2019

Revisions made on the manuscript to be submitted to Plos one

Health impact of hepatic-venous-occlusive disease in small town in Ethiopia – case study from Tahtay Koraro district in Tigray region, 2017 

Version: 1 Date: 26 September, 2019

Dear Editor-in-Chief:

Amendments made for the HVOD manuscript

Response to journal requirements’ comments

1. We have addressed the Plos one’s style requirement

2. Ethical review board had waived the requirement for informed consent.

3. The minor occurrence of overlapping text with previously published journals was rewritten and edited.

4. The data set is uploaded as supporting information file

5. We have referred the figures in our text

6. We have referred the table in our text

Response to reviewers’ comments

1. The Title was modified to Health impact of hepatic-venous-occlusive disease in small town in Ethiopia – case study from Tahtay Koraro district in Tigray region, 2017 as commented.

2. The result of abstract has included trend over time of VOD fatality.

3. In line 29 we have added the problem after however

4. Page 3 line 53 was modified to - in different literatures it has been reported that the incidence of VOD….

5. The comment given on page 4, line 69; We have added additional literatures concerned on hepatic-venous-occlusive disease in Tahtay koraro district

6. On page 4 line 72 we have removed the sentence according to the given comment

7. On page 4, line 76 we have included the population size of Tahtay Koraro district in 2017

8. In the result section we have edited each tables and figures by describing fully

9. The typographical errors were edited

---

## [Decision Letter · Decision Letter 1]

21 Oct 2019

Health impact of hepatic-venous-occlusive disease in small town in Ethiopia – case study from Tahtay Koraro district in Tigray region, 2017

PONE-D-19-19407R1

Dear Dr. Weldearegay,

We are pleased to inform you that your manuscript has been judged scientifically suitable for publication and will be formally accepted for publication once it complies with all outstanding technical requirements.

With kind regards,

Prof. Raffaele Serra, M.D., Ph.D

Academic Editor

PLOS ONE

Additional Editor Comments (optional):

amended manuscript is acceptable

Reviewers' comments:

Reviewer's Responses to Questions

**Comments to the Author**

1. If the authors have adequately addressed your comments raised in a previous round of review and you feel that this manuscript is now acceptable for publication, you may indicate that here to bypass the “Comments to the Author” section, enter your conflict of interest statement in the “Confidential to Editor” section, and submit your "Accept" recommendation.

Reviewer #1: All comments have been addressed

2. Is the manuscript technically sound, and do the data support the conclusions?

Reviewer #1: Yes

3. Has the statistical analysis been performed appropriately and rigorously? 

Reviewer #1: Yes

4. Have the authors made all data underlying the findings in their manuscript fully available?

Reviewer #1: Yes

5. Is the manuscript presented in an intelligible fashion and written in standard English?

Reviewer #1: No

6. Review Comments to the Author

Reviewer #1: The authors have responded to my comments. However, the manuscript still needs some editing. The organization of the results section as well as the flow of the paragraphs could be improved with some editorial help.

7. PLOS authors have the option to publish the peer review history of their article (what does this mean?). If published, this will include your full peer review and any attached files.

Reviewer #1: No

---

## [Editor Report · Acceptance letter]

24 Oct 2019

PONE-D-19-19407R1 

Health impact of hepatic-venous-occlusive disease in a small town in Ethiopia – case study from Tahtay Koraro district in Tigray region, 2017     

Dear Dr. Weldearegay:

I am pleased to inform you that your manuscript has been deemed suitable for publication in PLOS ONE. Congratulations! Your manuscript is now with our production department. 

With kind regards,

on behalf of

Prof. Raffaele Serra 

Academic Editor

PLOS ONE